# Lipid-Sensing Receptor FFAR4 Modulates Pulmonary Epithelial Homeostasis following Immunogenic Exposures Independently of the FFAR4 Ligand Docosahexaenoic Acid (DHA)

**DOI:** 10.3390/ijms24087072

**Published:** 2023-04-11

**Authors:** Stefanie N. Sveiven, Kyle Anesko, Joshua Morgan, Meera G. Nair, Tara M. Nordgren

**Affiliations:** 1Division of Biomedical Sciences, School of Medicine, University of California-Riverside, Riverside, CA 92521, USA; 2Department of Bioengineering, Bourns College of Engineering, University of California-Riverside, Riverside, CA 92521, USA; 3Department of Environmental and Radiological Health Sciences, Colorado State University, Fort Collins, CO 80523, USA

**Keywords:** FFAR4, GPR120, lung, inflammation, exposure, DHA

## Abstract

The role of pulmonary free fatty acid receptor 4 (FFAR4) is not fully elucidated and we aimed to clarify the impact of FFAR4 on the pulmonary immune response and return to homeostasis. We employed a known high-risk human pulmonary immunogenic exposure to extracts of dust from swine confinement facilities (DE). WT and *Ffar4*-null mice were repetitively exposed to DE via intranasal instillation and supplemented with docosahexaenoic acid (DHA) by oral gavage. We sought to understand if previous findings of DHA-mediated attenuation of the DE-induced inflammatory response are FFAR4-dependent. We identified that DHA mediates anti-inflammatory effects independent of FFAR4 expression, and that DE-exposed mice lacking FFAR4 had reduced immune cells in the airways, epithelial dysplasia, and impaired pulmonary barrier integrity. Analysis of transcripts using an immunology gene expression panel revealed a role for FFAR4 in lungs related to innate immune initiation of inflammation, cytoprotection, and immune cell migration. Ultimately, the presence of FFAR4 in the lung may regulate cell survival and repair following immune injury, suggestive of potential therapeutic directions for pulmonary disease.

## 1. Introduction

Lipids play a critical role in metabolic and immune homeostasis, and this underscores their potential as diet-based therapeutics in the treatment of human disease. One such family of lipids with well-characterized anti-inflammatory and pro-resolving effects includes omega-3-polyunsaturated fatty acids (ω-3-PUFA) such as docosahexaenoic acid (DHA) [1,2,3]. Dietary supplementation with PUFA introduces these lipids into lipoproteins, which can be cleaved by lipoprotein lipase in the vasculature and target tissues. Use of ω-3-PUFA, experimentally and clinically, demonstrates improvements in the resolution of inflammation, tissue repair, cardiovascular health, metabolic homeostasis and dyslipidemia, and chronic diseases such as chronic obstructive pulmonary disease (COPD), among others [2,4,5,6,7,8,9,10,11]. Among these ω-3-PUFAs, studies identify that DHA, in particular, exhibits the broadest anti-inflammatory response via numerous modes of action and thus has strong therapeutic potential in diseases such as those of the lung [12,13,14]. DHA is critical to cell membrane integrity and participates in anti-inflammatory signaling through the generation of lipid mediators and through the ligation of lipid-sensing free fatty acid receptor 4 (FFAR4) [15,16,17,18]. 

FFAR4, previously GPR120, is highly expressed in the gut, where it is activated by an ingested luminal long-chain fatty acid, resulting in energy homeostasis regulation. FFAR4 is targeted therapeutically in clinical trials for the treatment of metabolic disease (clinicaltrials.gov identifier: NCT03285750, NCT02444910, NCT03062592, NCT05068557). However, this lipid-sensing role of FFAR4 is less intuitive in the pulmonary system, which exhibits some of the highest *Ffar4* expression [19,20]. Publications outlining the functionality of FFAR4-signaling in the pulmonary context are limited to a handful of papers, but, importantly, a role in improved airway repair in a murine model of naphthalene-induced airway injury has been elucidated by Lee et. al. [21]. We have previously reported the benefits of DHA administration on inflammatory outcomes, both in vitro and in vivo, using an organic swine dust exposure model which causes pulmonary inflammation [1,2]. These data suggest potential crosstalk between DHA and FFAR4 in protecting the lung, so these may be important as therapeutic targets in lung tissue repair [22]. 

This model is of clinical importance due to the Centers for Disease Control’s classification of agricultural dust inhalation as one of the highest-risk occupational exposures, as it contains organic and inorganic immunogenic particulates (bacteria, viruses, fungi, and chemicals) that drive lung inflammation [23,24,25,26]. The long-term implications of these exposures, which are typically frequent and repetitive, are an increased risk of pulmonary inflammatory diseases such as COPD, the third leading global cause of death per the World Health Organization [27,28]. COPD is marked by two overlaying, irreversible pathologies: chronic bronchitis and emphysema, which often result from inhaled exposures and dysregulated repair. Without established therapeutics to reverse lung damage, exploring DHA and FFAR4 as potential targets in lung repair is enticing, especially as GPCRs comprise over 40% of current therapeutic targets and data demonstrates the positive role of DHA on COPD morbidity [29,30]. 

Due to the numerous ways DHA functions in vivo, these anti-inflammatory and pro-repair effects are not always FFAR4 dependent [31,32]. Thus, this current work aims to identify the role of FFAR4 in our previous elucidation of the DHA-mediated resolution of pulmonary inflammation. To accomplish this aim, we employed a well-established mouse model of agricultural dust-induced lung inflammation and complemented these studies using additional exposure models known to facilitate lung damage through different mechanisms. Our mouse model of this dust exposure allows for the study of pulmonary inflammation and resolution, as these exposure models are self-limiting [33,34]. This study clarifies the impact of FFAR4 on the inflammatory process through repair, with the long-term objective of ultimately identifying potential therapeutics, such as against DHA and FFAR4, and interventions for chronic lung diseases, for which few current therapeutics are available. 

## 2. Results

### 2.1. Ffar4 Deficiency Leads to Dampened Inflammatory Cell Influx to the Airways Independently of DHA

*Ffar4* wild type (WT), heterozygous (HET), and knockout (KO) mice were supplemented with DHA or mineral oil vehicle by oral gavage and repetitively exposed to hog barn dust extract (DE) or saline vehicle (Figure 1a). Total immune cell influx to the airways was quantified from bronchoalveolar lavage fluid (BALF). As previously published, exposure to DE significantly induced the recruitment of cells into the airway compartment in WT mice, hallmarked by neutrophilia in BALF (Figure 1b–e). Notably, cell influx following DE exposure was significantly diminished in *Ffar4* KO mice compared with WT, independently of DHA administration (Figure 1c; *p* < 0.0001). Analysis by 3-way ANOVA revealed significant main effects of genotype, treatment, and exposure. Further, Tukey’s multiple comparisons calculated a significantly higher total cell count in WT than in KO in the vehicle supplement group (Figure 1c,d). Though total cell counts were altered, there was no significant change to cell differentials apart from a significant induction of neutrophils in dust-exposed airways compared with genotype-matched saline controls (Figure 1e; main effect of infection: *p* < 0.0001). To address differences in airway compartment inflammation, perivascular inflammation was scored from hematoxylin and eosin (H & E)-stained formalin-fixed, paraffin-embedded (FFPE) tissues using our previously published histopathology scoring techniques [35]. Dust exposure significantly induced perivascular inflammation with immune cell infiltration in both WT and KO mice compared with their genotype-matched saline controls, while DE-exposed KO had significantly higher perivascular inflammation scores compared with DE-exposed WT mice (Figure 1f; Adj. *p* = 0.0127 and <0.0001 respectively).

We additionally challenged these mice with an exposure eliciting a differently skewed inflammatory response of helminth infection with the rodent hookworm Nippostrongylus brasiliensis (Nb), which infects the lung and small intestine (Figure 2a). The subsequent type-2 skewed immune response and lung epithelial damage causes M2-like (alternatively activated) macrophage polarization, which contrasts with the M1-like macrophage polarization following DE exposure [36]. Like the DE model, *Ffar4*-deficient mice had significantly less immune cell recruitment to the airway following helminth infection (Figure 2b; Adj. *p* = 0.0294). Similarly as well to the BAL data in the DE model, the infection promoted eosinophilia, as expected, in both genotypes (main effect of infection: *p* ≤ 0.0001), while the proportions of immune cells implicated in helminth infections (alveolar macrophages and eosinophils) were equivalent (Figure 2d,e). Despite decreased airway cell recruitment, *Ffar4*-deficient mice exhibited improved anti-helminthic immunity, with significantly reduced total worms counted in the intestines (Figure 2f; *p* = 0.0006). Together, these data reveal an important role of FFAR4 in airway total immune cell counts with no identified changes in present populations during immunogenic exposures, regardless of whether the stimulus induces a neutrophilic or eosinophilic response.

### 2.2. Dysregulated Epithelial Homeostasis Is Exacerbated by DE Exposure in Ffar4-Null Mice

To identify if DE-exposed mice had epithelial pathology, histopathological scoring was conducted on H & E-stained lung tissue from mice that had been repetitively exposed to DE. Overall, *Ffar4*-deficient lung tissue had significantly dysplastic epithelium compared with WT mice (Figure 3b,c; Adj. *p* < 0.0001). The average thickness of the epithelium of respiratory airways was significantly increased in DE-exposed *Ffar4*-deficient lung tissue compared with those from DE-exposed WT and naïve KO mice (Figure 3d; Adj. *p* = 0.0001 and 0.0002 respectively). DE exposure significantly altered the airway epithelial area in WT and KO mice compared with their genotype-matched saline controls, with no significant differences determined between genotypes (Figure 3e; Adj. *p* = 0.0285 and 0.004, respectively). To assess if the self-limiting nature of this known epithelial disruptor is FFAR4-dependent, mice repetitively exposed to DE were given a 3-dayexposure-free recovery period before lung barrier integrity was assayed using a commonly employed epithelial permeability assessment with 70,000 KDa Rhodamine-B-conjugated dextran (RhoB), which was administered intranasally to assess pulmonary permeability [37]. WT mice showed no difference in plasma RhoB levels between DE-exposed and saline control mice, while DE-exposed *Ffar4*-null mice had significantly elevated plasma fluorescence compared with DE-exposed WT mice (Figure 3f; Adj. *p* = 0.0081). To supplement this functional data, we used a model of acute lung injury to directly challenge the epithelial response to chemical injury. Porcine pancreatic elastase (PPE) or vehicle saline was administered intranasally at 0.9–1.2 U. *Ffar4*-KO mice were significantly less likely to survive this acute epithelial injury model (Figure 3h, Log-rank test: *p* = 0.0224). 

### 2.3. Hippo Pathway as a Potential Mechanism of Dysregulated Immune Recruitment and Epithelial Homeostasis

Among a multitude of diverse functions, the Hippo pathway regulates the stability of the transcriptional co-activators yes-associated protein 1 (YAP) and transcriptional coactivator with PDZ-binding motif (WWTR1/TAZ). This well-conserved pathway is an important and well-established regulator of both lung epithelial homeostasis and the immune system [38]. In mammals, YAP signaling downstream of Hippo promotes inflammatory responses, including the bacterial response which dominates following swine dust exposure [39]. Further, GPCRs which signal through Gαq11 and tight junction proteins are established regulators of this pathway. Indeed, FFAR4 has been implicated as a regulator of the Hippo pathway in some cancers; however, there is little in the literature regarding this synergism in the lung [39,40]. Given the importance of YAP/TAZ signaling in lung epithelial homeostasis, the significant epithelial dysplasia with barrier and immune-response deficits in *Ffar4*-deficient mice led to the hypothesis of YAP/TAZ dysregulation in mice lacking *Ffar4* [41,42]. To investigate this, lung sections were taken from FFAR4−/− mice and wild-type littermates after exposure to 12.5% DE or saline control as described above. YAP and TAZ staining intensity was assessed semi-quantitatively across complete sections (Figure 4), revealing a significant reduction in YAP intensity in the lungs of *Ffar4*−/− mice (2-way ANOVA; main effect of genotype, *p* = 0.0424). These findings are consistent with the impaired immune response, epithelial dysplasia, and barrier dysfunction, suggesting a potential role of the Hippo pathway and/or the effector YAP in FFAR4-mediated epithelial homeostasis.

### 2.4. Genes Related to the Innate Immune Response and Apoptosis Are Downregulated in Ffar4-Deficient Mice

We sought to examine gene expression changes that could corroborate the DHA-independent phenotypes (i.e., epithelial dysfunction, dampened immune infiltrate, and reduced YAP intensity by IF) from repetitive DE exposure in *Ffar4* WT and KO mice using total lung RNA from vehicle-treated WT and KO mice repetitively exposed to saline or DE. As expected, both wild-type and *Ffar4*-null mice exhibited upregulation of numerous immune-related genes following repetitive DE exposure in the inflammatory model (Figure 5a,b). DE exposure significantly induced 51 differentially expressed genes (DEG) in the WT comparison and 34 in the KO comparison (Figure 5a,b, respectively; Adj. *p* < 0.05). 

Most genes were upregulated following DE exposure, though there were a few genes downregulated in DE compared with saline groups (five genes in WT and one in KO). The five DEGs which were downregulated in WT following DE-exposure include immunomodulatory genes (Ackr4, Ahr, Gata3), while Adgre5 affects cell–cell adhesions and Phlpp1 regulates mucus production. The downregulated gene from the KO comparison group is Nox4, which plays a critical role in the generation of reactive oxygen species (ROS) from polymorphonuclear neutrophils, important in the progression of inflammation, particularly in this neutrophilic dust-exposure response.

Of these DEGs derived from each genotype comparison of DE-exposed to genotype-matched saline controls, 14 upregulated DEGs were common between WT and KO comparisons (Figure 5c). These 14 common DE-induced DEGs exhibit functions such as chemokine and interleukin signaling (e.g., Ccl20, Cxcl3, Cxcl13, Il1r2, Tlr2); LPS detection and signaling cascades (e.g., Cd14, Tlr2); and TLR4 signaling (e.g., Tlr2, Cd14, Itgam), among others. Comparing the genotype-specific responses in DE exposure, *Ffar4*-deficient mice had five downregulated DEG compared with WT and one upregulated gene (Figure 5d). Uniquely expressed DEGs by genotype were grouped according to pathway analyses to determine genotype-driven functional differences in the responses to dust (Table 1). DE exposure in KO mice alters pathways involved in pyroptosis and necrosis, which could implicate a role for FFAR4 in cell survival (Table 1). The pathway analysis from the genotype-specific responses reveals effectors of toll-like receptor (TLR) signaling cascades among the genes downregulated in KO lungs. Overall, the gene expression differences in dust-exposed WT and *Ffar4*−/− mice allude to a deficiency in key players of the innate immune response and differential responses to DE exposure.

## 3. Discussion

Lung tissue is an abundant expresser of the lipid-sensing GPCR and FFAR4 molecules, canonically involved in metabolic regulation and insulin sensitivity, but which also play a role in PUFA-mediated anti-inflammation [3]. PUFAs, like the omega-3 fatty acid DHA, are well-established molecules in attenuating inflammation, tipping the balance towards the resolution of the inflammatory process. Our previous data have revealed that DHA dampens the inflammatory response to swine dust exposure, lessening the release of protein inflammatory mediators as well as the recruitment of inflammatory cells, namely neutrophils, to the lung [1,2,43]. The high expression of FFAR4 in the lung, and the fact that dietary lipids are first pumped through the pulmonary system before traveling to the liver, led us to hypothesize that DHA ligation of pulmonary FFAR4 is important in the observed dietary-DHA-attenuated pulmonary inflammation. 

As previously characterized, DE exposure induced BAL neutrophilia independently of the FFAR4 status, and there were no significant differences in the recruited cell types determined from cytospins (Figure 1c–e) [44,45,46,47,48]. While DHA did significantly dampen the dust-induced airway immune cell recruitment in wild-type mice within these current studies (*p* < 0.0001), *Ffar4*-deficient mice had reduced total immune recruitment, independent of DHA administration, which were similar to saline controls (Figure 1b,d). It has been discussed in the literature that DHA can mediate anti-inflammatory effects independently of FFAR4-signaling, the means of which include bioavailability of immune-related substrates, lipid raft modifications which interrupt immune signaling pathways, and DHA-derived mediators called specialized pro-resolving mediators (SPMs), such as resolvins [32,49,50,51]. 

These studies showing that treatment with DHA alleviated the inflammatory cell influx are in line with what we have previously published. Surprisingly, FFAR4 had a significant and independent effect on cell influx (Figure 1d) [1,2]. Entrapment of immune cells in the mesenchyme has been demonstrated in *Myd88*-null mice repetitively exposed to DE [33,43]. These published studies revealed no significant changes to cells in the BAL but with significant histopathologic changes in the *Myd88*-deficient mice, including immune cell entrapment and epithelial dysplasia. Similar outcomes were significantly altered in *Ffar4*-null mice in our studies, which also showed significantly increased perivascular inflammation scores using blinded histopathology scoring (Figure 1f). FFAR4 may regulate the migration of immune cells out of the vasculature and into the airway. Thus, we complemented these findings using an eosinophilic model of inflammation to evaluate the consistency of these data across different immune responses [52,53]. Here, *Nippostrongylus brasiliensis* (Nb) infection was utilized to identify if the observed deficiency in FFAR4 with DE exposure was inclusive of other exposures promoting airway immune infiltration. Similarly, this eosinophil-skewed parasitic response, which significantly elevated BAL cell counts in wild-type mice (*p* = 0.0060), was also dampened by the loss of *Ffar4* without changes to recruited cell populations (Figure 2a–c, respectively).

These findings corroborated the dysregulated immune cell presence in the airway as seen in the DE exposure model, regardless of the type of immune response elicited. What these data suggest is that a lack of *Ffar4* plays a role in the migration of immune cells into the airway space. The expectation of anti-helminthic immunity, if simply considering the impaired immune recruitment, would be increased parasite survival in mice lacking *Ffar4*. Interestingly, the parasite burden was significantly lower in *Ffar4*-deficient mice (Figure 2f), however, the lack of *Ffar4* in the gut may differentially impact parasite survival in the jejunum, from which the worm burdens are quantified [54]. This is a limitation of studies utilizing global gene knockout strategies; however, removing *Ffar4* specifically from the lung would be challenging since it is expressed across a number of lung-resident cells (i.e., pneumocytes, endotheliocytes, macrophages etc.), and it would require a number of cell-specific knockout mouse strains to identify which cells contribute to each phenotype.

From the blinded histopathological scoring of lung epithelium, we identified significantly increased dysplastic appearance and airway epithelial thickening, resembling hyperplasia, in DE-exposed *Ffar4*-null airways compared with wild-type (Figure 3b–d; *p* = 0.0079 and *p* = 0.0079, respectively). It has been demonstrated that silencing FFAR4 in intestinal cell lines results in increased cell proliferation [55]. The epithelial dysplasia phenotype from the histopathological analyses in our *Ffar4*-deficient mice could similarly be the result of increased proliferation of lung epithelial cells upon immunogenic activation, and more studies to examine this potential are warranted. The literature dictates that FFAR4 signaling dampens cell proliferation and maintains mucosal barriers; thus, mice deficient in *Ffar4* exhibit lung epithelial dysplasia possibly through unregulated cell proliferation, which could further translate to impaired barrier integrity [55,56]. Of note, epithelial dysplasia (usually hyperplasia of certain cell populations) and increased permeability are identified in epithelial pathologic remodeling among individuals with COPD, a disease for which hog barn dust exposure is a risk factor [24,57,58,59]. 

We sought to identify if the FFAR4-dependent epithelial dysplasia had functional consequences for barrier integrity. Exposure to airway immunogens such as diesel exhaust, allergens, and environmental dust results in mucosal epithelial barrier dysfunction marked by increased permeability, but this is reversible upon successful epithelial repair mechanisms [60,61]. Further, in vitro studies using human bronchial epithelial cells, grown in an air-liquid interface, demonstrate increased dextran permeability following repetitive swine dust extract exposure [62]. Unlike in cases of human exposure, which are linked to chronic pulmonary deficits, the DE dose used in our mouse exposure model is self-limiting by design and the lungs can recover under homeostatic conditions. Thus, we sought to clarify if this return to homeostasis is *Ffar4*-dependent, particularly in the epithelial restoration of barrier function. In a pilot in vivo study, we identified that repetitive DE-exposure increased pulmonary epithelial permeability to intranasally delivered 70,000 KDa Rhodamine-B-conjugated dextran (RhoB70) (Appendix A). Similarly, previous data using intestinal epithelium have demonstrated that Villin-cre *Ffar4*-floxed mice exhibited increased intestinal permeability and decreased expression of basement membrane-affiliated genes, suggestive of the role of FFAR4 in mucosal barrier integrity [55]. After implementing a three-day recovery period following repeated DE exposures, we identified persistent barrier deficits among *Ffar4*-null mice that were not identified in the wild-type mice which restored barrier integrity by this time. Mice lacking *Ffar4* had increased leakiness to RhoB70, measured as plasma fluorescence (Figure 3f; *p* = 0.0368). Like data reported in the literature, *Ffar4*-expression in these mice impacts the efficient reestablishment of epithelial barrier integrity during this early time point of repair in our DE model. Further, tight-junction proteins are known to mediate the transmigration of immune cells, and dysplastic epithelium with poorly assembled tight junctions could prevent the transepithelial migration to the airway compartment following immunogenic challenge [63]. This could be a contributing factor to the decreased cell counts in BAL, which contrasted with the histopathologic findings of increased immune infiltrates in the mesenchymal lung compartments in mice lacking *Ffar4* (Figure 3c,d). 

We then assessed the functionality of the epithelium in an acute setting using a porcine pancreatic elastase (PPE)-induced acute lung injury model. Mice partially or completely deficient in *Ffar4* were less likely to survival during a 48 h timepoint while WT mice had 100% survival (Figure 3g–h). This data supports the hypothesis of insufficient inflammatory activation and/or impaired epithelial integrity in the absence of *Ffar4* which, in the case of PPE injury, resulted in death [64]. The luminal and basilar polarization of lung epithelium is imperative to proper epithelial receptor signaling of membrane-associated receptors, e.g., through GPCRs such as FFAR4. The significant epithelial dysplasia and subsequent impaired polarization of these cells could contribute to inadequate membrane-receptor-mediated responses to stimuli and impaired immune cell interactions during migration to the airway. To explore a potential link to these mechanisms, we turned to the Hippo pathway. This pathway regulates cell proliferation and survival and is important in modulating pulmonary epithelial homeostasis. GPCRs, and particularly those signaling via Gαq11, such as FFAR4, are known regulators of the Hippo pathway [65,66]. 

Relevant to the DE-exposure model, TLR4 activation is upstream of the Hippo pathway, and LPS has been shown to activate actuators of the Hippo pathway. Thus, the Hippo pathway plays a role in mediating TLR4-dependent immune response through lymphocyte trafficking, antigen recognition, as well as immune tolerance [39]. With these considerations in mind, we performed immunofluorescence labeling of the Hippo actuators YAP and TAZ on FFPE lung sections following repetitive dust exposure. A *Ffar4*-dependent decrease in YAP activity was quantified, which complements the epithelial deficits in this work based on the published role of decreased YAP leading to impaired epithelial repair (*p* = 0.0427, Figure 5) [42,67]. The decreased expression of YAP may be linked to the observed epithelial dysregulation or the dampened immune response measured as recruited cells to the airway. Given the apparent role of YAP in the lung injury response, decreased YAP expression may also contribute to the epithelial healing defects observed in *Ffar4*^−/−^ mice; further work is needed to elucidate these mechanisms.

We next aimed to clarify potential gene signatures underpinning the immune and epithelial deficits identified in the absence of FFAR4 using an RNA panel tailored to immunological responses. These data provided insight into DE exposure- or genotype-related transcript alterations. Total lung tissue RNA from repetitively exposed wild-type and *Ffar4*-null mice revealed that DE exposure, when compared to genotype-matched saline controls, resulted in the upregulation of several of the genes assayed, with far fewer downregulated genes. This is in line with expectations that an immunogenic stimulus, such as the swine dust extract, would promote the upregulation of inflammation and immune-related genes [68]. DE exposure in WT and KO mice resulted in 46 and 33 upregulated differentially expressed genes (DEGs), respectively. Fourteen DEGs are shared between DE-exposed KO and WT samples, with upregulation of genes important in innate-immune recognition of DE components (*Cd14, Tlr2, Pigr*) and immune cell chemoattraction (*Cxcl13, Ccl9, Ccl20, Il1r2, Itgam*).

Significant pathways associated with these upregulated genes include arms of the innate and adaptive immune response, which occurs as the inflammatory response to repetitive exposure connects innate and adaptive immunity. The innate immunity pathways in DE-exposed WT mice include ‘complement activation’, while the adaptive pathways include the regulation of ‘control Treg development’ and ‘IL-4/IL-13/IL-10 signaling’. These phenomena in repetitive swine dust exposure are discussed in the literature from both in vitro and in vivo studies [34,69,70,71,72]. The only overlapping pathway term in DE-exposed KO data is ‘IL-10 signaling’, suggesting a transition from the acute response to chronic/adaptive response to DE mediating immune-regulatory signaling to control inflammation. The KO DE response is marked by pathways involving ‘inflammasomes’, ‘pyroptosis’, ‘immune cell migration’, ‘necrosis’, and ‘IL-1 signaling’. Inflammasome pathways, pyroptosis and necrosis are all related to cell death, suggesting a cytoprotective role of *Ffar4* in the inflamed lung, a role that has not been well characterized in publications. However, DHA-mediated cytoprotection and redox regulation has been explored and determined to have some dependence on FFAR4 [73,74,75,76]. These pathways may provide context for the lack of cell infiltration in the BAL, barrier deficits, and poor survival to acute lung injury among *Ffar4*-null mice in this work. 

Another major difference in DE-exposed outcomes is the activation of complement pathways in WT but not KO mice. Complement activation is a characteristic response to swine dust exposure in humans, both in vivo and in vitro [77]. Complement is an essential part of pathogen recognition and plays an important role in both the induction of sufficient immune responses and pathogen clearance for adequate resolution [78,79]. Of the five DEGs comparing KO with WT DE-exposed mice, an important complement effector gene, *Cfb*, was downregulated in mice lacking *Ffar4*. This may contribute to a failure to sufficiently promote the alternative complement pathway and initiate a sufficient inflammatory response, seen as dampened immune infiltration in the BAL. In addition, lower expression of *Irf7, H2k1, Ly96,* and *Cxcl13* will also dampen the initial and sustained immune response following DE exposure. It has been shown that LPS-challenged mice exhibit an interferon response in an IRF7–TLR4-dependent manner, with deficiency leading to dampened cytokine release [80]. *H2k1* is critical in antigen processing and presentation via major histocompatibility receptor I (MHCI) in the progression of the inflammatory response. In addition, epithelial progenitor populations in the lung also express *H2k1* and drive barrier restoration [81]. Lymphocyte antigen 96 (*Ly96*), also called myeloid differentiation factor 2 (Md2), dimerizes with TLR4 to recognize LPS and initiate MyD88 signaling cascades driving pro-inflammatory gene expression in response to swine dust exposure [33,82,83]. MyD88 KO mice repetitively exposed to DE exhibited significantly reduced chemokine and cytokine release, dampened BAL cell influx, and epithelial dysplasia. Though not as pathological a phenotype as in the MyD88 KO, this reduced *Md2* and *Irf7* expression could contribute to MyD88-dependent dampened BAL infiltration and epithelial dysplasia in the *Ffar4*-null mice from our studies. Taken together, these data reveal phenotypic findings such as immune response deficits, failure to survive, and impaired epithelial homeostasis which are corroborated by gene expression data in mice lacking *Ffar4*. 

These results broaden the scope for FFAR4-signaling and pose new questions as to its role in lung homeostasis. Chronic inflammatory diseases, such as COPD and diabetes mellitus, are hallmarked by altered lipoprotein profiles and lipid metabolism disorders causing cardiovascular comorbidities [84,85]. A high-fat diet and diabetic dyslipidemia induces atherosclerosis and lipoprotein imbalances in human and non-human studies [86,87]. DHA is recommended clinically for dyslipidemia because it attenuates atherogenic lipoproteins and FFAR4 senses excess lipids in high-fat diets [5,88]. Studies have shown a decrease in FFAR4 during heart failure, with *Ffar4*-null mice exhibiting aberrant oxylipin transcriptomics during cardiac pressure overload. FFAR4-signaling was responsible for the maintenance of oxylipins in lipoproteins and may be a novel therapeutic strategy in the treatment of atherosclerosis [89]. Clinical trials targeting FFAR4 are abundant in the treatment of diabetes-related metabolic disorders, highlighting the potential diverse utility of this receptor in the treatment of lipid- and inflammatory-related diseases [90,91]. The results of these studies must be taken within the context of global *Ffar4* knockout mice, as it cannot be ignored that many of these effects may be the result of systemic metabolic changes in these mice. DHA as well as FFAR4 signaling regulate the availability of metabolites and lipid mediators that could impact the balance in the immune system and at the cellular level. This work underscores the need to study non-canonical roles of FFAR4-signaling on innate immune responses at mucosal barriers where *Ffar4* is highly expressed. Future directions for this work include determining the pathway effectors for this FFAR4-mediated regulation of pulmonary immunity and epithelial homeostasis in order to identify the potential of FFAR4 therapeutics in the treatment of lung disease. 

## 4. Materials and Methods

### 4.1. Ffar4 Knockout Mouse Model

The knockout mouse model used in these studies was generated by Bjursell et al. [31]. As described in their publication, the *Ffar4* gene was removed and replaced by a LacZ reporter sequence in C57BL/6NCrl (Charles River) mice. Animals were housed in a specific pathogen-free facility (12 h dark, 12 h light) with ad libitum access to standard chow. Heterozygous mice were used as breeders, allowing for the comparison of littermates in these studies. Mice were used between 8–12 weeks of age and experiments were conducted with age, sex, and littermate-matched controls.

### 4.2. 3R Statement

Mice were bred for use in specific experiments and multiple tissues were collected from experimental mice to reduce the number of animals used in these studies. All mice euthanized were used for experimental or training purposes to ensure the efficient and limited use of mice. All experiments were performed with approval from the University of California (Riverside, CA, USA) Animal Care and Use Committee (A-20210017; and A-20200014), in compliance with the US Department of Health and Human Services Guide for the Care and Use of Laboratory Animals.

### 4.3. Preparation of Hog Barn Dust Extract (DE)

Settled dust was collectedfrom at a height of approximately 1 m from within confined animal feeding facilities housing 500–700 swine, as previously described [92,93]. Collected dust samples were stored at −20 °C, and extracts were periodically generated and stored as 100% stock. To prepare the 100% extracts, 5 g of settled dust was stirred into 50 mL of sterile Hank’s balanced salt solution for 1 h at room temperature in a fume hood. The stirred solution was then transferred to a polypropylene 50 mL conical tube, centrifuged at 2500 rpm for 20 min at 4 °C. This step was performed twice, collecting only supernatant fractions after each centrifugation step. The supernatant fraction was then sterile-filtered using 0.22 μm syringe filters and aliquoted into 1.5 mL microcentrifuge tubes. For mouse experiments, 12.5% DE was prepared from the 100% solution by dilution into sterile phosphate-buffered saline (PBS) which was stored at −20 °C and thawed just before use. All dust extracts were generated from the same settled dust collection; therefore, no batch effects are present.

### 4.4. Dust Exposure Model with Exogenous DHA Administration

This model has been utilized and previously published to study the impacts of inhaled exposure on murine respiratory responses [93]. DHA was administered as previously published; DHA (Cayman Chemical, Ann Arbor, MI, USA) stock solution was prepared in mineral oil at 20 mg/Ml [1]. Each mouse received a 2 mg dose of DHA, delivered in a 100 μL bolus by oral gavage, daily for 12 consecutive days (6 days prior to DE exposure and 7 days concurrently). Following oral gavage on days 7+, mice were anesthetized under isoflurane using a small-animal anesthesia vaporizer set to 1.5–2% (*v*/*v*). Mice were determined to be anesthetized when there was a noticeable slowing of their breathing rate and a negative hind pedal reflex. Using a micropipette, 50 μL of well-mixed 12.5% DE or PBS was administered over both nares until fully inspired. Animals were held supine and angled upright for a few seconds and replaced in the cage as they began to awaken. The DE was administered once daily for 7 consecutive days, and animals were sacrificed 5 h following the final DE dose. For the recovery studies, mice were not administered DHA or vehicle and were instead given a 3-day recovery period following the final dose of the 7-day DE exposure period. Mice in the recovery group were intranasally administered fluorescence-conjugated dextran, FITC 4 KDa (Sigma Aldrich, cat no. 46944, St. Louis, MO, USA), and Rhodamine B 70 KDa (Sigma Aldrich, cat no. R9379 at a concentration of 8 mg/kg of each conjugate, q.s. to 50 μL in sterile PBS. The dextran was prepared individually for each animal immediately before instillation and the animals were euthanized 1 h after dextran administration. To achieve 85% power in detecting a 20% difference in histopathological changes, the least sensitive parameter in quantifying DE-induced outcomes, our power analyses determined 8 mice per group were needed in these studies. Experiments were repeated 3–4 times with at least 3 animals from each group represented in every experiment, apart from elastase studies which were repeated twice.

### 4.5. Acute Lung Injury Model

Intranasal delivery of porcine pancreatic elastase (PPE) has been used to model acute lung injury and emphysema [94,95]. For these studies, mice were given a single instillation of 0.9 or 1.2 U PPE (Sigma cat. No. E7885) in PBS at 50 μL total volume. 

### 4.6. Parasitic Worm Infection Model

*Nippostrongylus brasiliensis* life cycle was maintained via rat infection and collected feces were cultured in vermiculture medium on humidified petri dishes. L3 larvae were gravity-extracted in saline from these plates on the day of infection. Mice were anesthetized by isoflurane and subcutaneously injected with 500–600 L3 worms in 200 µL PBS or PBS alone using a 22 G needle (considered infection day 0), as previously published [53,96]. Mice were euthanized for tissue collection on day 7 post infection.

### 4.7. In Vivo Outcomes

For in vivo outcomes, the experimentalist was blinded to the genotype and treatment of the mice. 

*Plasma*: Upon euthanasia of the mice, 250–500 μL of blood was collected from the renal artery using a 25-gauge needle and transferred to a lavender-capped blood collection tube. The blood was centrifuged for 15 min at 2000× *g* and plasma was transferred to a microcentrifuge tube and stored at −80 °C. 

*Bronchoalveolar lavage (BAL):* A cannula was placed into the trachea, near the hyoid, and tied off with suture string to avoid movement (BD, cat no. 381434). For BAL washes, 1 mL of ice-cold PBS was slowly administered through the cannula and slowly withdrawn. The first wash was reserved for cytokine analyses by enzyme-linked immunoassays (ELISA), while the remaining two washes were stored together in a separate tube. All tubes were centrifuged at 1200 rpm for 5 min at 4 °C to pellet the airway cells, and the supernatant fraction of the first tube was aliquoted to store at −80 °C for use in the ELISA. All three cell pellets were combined and treated with red blood cell lysis buffer. A total cell count was generated manually by a hemocytometer using masked sample IDs, counted twice, and averaged on the day of euthanasia. Cytospins were prepared from 100,000 cells for each BAL sample and were stained by Diff-Quick and coverslipped with a toluene-based mounting medium. The cytospins were imaged (5 images per slide), and a square box of the exact same size was centered on each image. A total of 300 cells were counted from within the boxes and marked as either macrophages, neutrophils, eosinophils, or lymphocytes to determine cell differentials of the airway cells. 

*RNA*: Following BAL fluid collection, the left main bronchus was tied off with a suture and the left lung was cut away, placed in a 1.5 mL microcentrifuge tube and flash-frozen in liquid nitrogen. The frozen lung was crushed into a powder using a liquid-nitrogen-cooled mortar and pestle (BelArt Cat. no. H37260-0100). Samples of 100 μg of tissue powder were used for RNA isolation with commercial RNA isolation kits (Invitrogen Purelink). 

*Lung histology*: The right lung and trachea were excised, inflated with 400 μL of 10% buffered formalin, and transferred to an apparatus to finalize the inflation of the lungs at constant pressure (20 cm). The lungs were trimmed and transferred to cassettes stored in EtOH. The cassettes were shipped to the University of California Irvine (UCI) Experimental Tissue Resource facility where they were paraffin-embedded. We obtained sections that were H&E-stained by UCI for histopathological analyses.

### 4.8. NanoString nCounter Transcript Analyses

RNA was prepared from flash-frozen, powdered left lung using commercial RNA kits as described above. Initial concentration and purity were determined using a Nanodrop instrument. RNA was cleaned up with additional on-column washes when contamination persisted. A second concentration was determined using Qubit and then samples were delivered to the Genomic Core Facility at UC Riverside for bioanalyzer analyses for sample QC prior to nCounter® analysis. Qubit concentration values were used to determine nCounter® panel load volumes, adjusting for poorer quality RNA as determined by the bioanalyzer. Samples were prepared with the Mouse Immunology V2 code set from NanoString per the manufacturer’s protocols, hybridized for 16 h, and left in the final 4 C cycle for no longer than 30 min. The nCounter^®^ runs were completed per manufacturer instructions using an nCounter^®^ SPRINT Profiler, with an N of at least 3 per group and as previously published [97]. Normalization was completed using nSolver 4.0 Data Analysis software with default settings, except that the background thresholding was set to 30 for these analyses. Heatmaps were generated using the ROSALIND™ online analysis platform (www.rosalind.bio, accessed on 6 February 2023) with gene cutoffs set to *p* < 0.05 and fold-change of + or −1.5 times. Differentially regulated genes that were statistically significant were input into REACTOME (www.reactome.org, accessed on 6 February 2023), an open-source pathway database, for gene pathway analyses [98,99]. Only pathways with a *p* < 0.05 and FDR < 0.1 were reported. 

### 4.9. Histopathological Scoring of FFPE lungs

*Airway dysplasia scoring*: Hematoxylin and eosin-stained slides were blinded at the start of the scoring period. Slides were first scanned on 4× and 20×, and notes were taken on each of the lung compartments (pleura, conducting airway, alveolar, and vascular compartments). Prominent changes to airways were noted and airways with complete ring-cross-sectional structures were analyzed to avoid any artifact of sagittal airway cuts and branching. Published examples of airway dysplasia were referenced and images were taken from the sample slides to establish an in-house reference of scores from 0–5 [100]. Slides were then scanned after establishing the scoring criteria, and a score was given to 5–7 individual airways per slide and averaged for the final dysplasia score.

*Airway area calculations*: Blinded images were taken at 20× of airways that were complete (forming a circle) to be used in these analyses. The basement membrane border was traced in ImageJ to represent the outer airway circumference [101]. The total area measurement was taken, and the luminal circumference was selected to give luminal area measurements. Subtracting the luminal area from the total area gave the representative epithelial area. Using the outer area, we calculated the predicted diameter to normalize the epithelial area to airway diameter. 

*Airway thickness calculations*: From the same blinded images, the thickness was measured as a linear length in ImageJ at three randomly selected points of each airway. These points were averaged across 5–7 airways to generate an average airway thickness for each sample. 

*Perivascular inflammation scoring*: Blinded slides were scanned for vascular inflammation and the presence of immune cell aggregates and diffusion surrounding medium to large vessels. The mice were not perfused, so vessels were easily identified by the presence of erythrocytes and surrounding smooth muscle cells. As before, images were taken to establish a scoring reference from 0–5 based on the thickness of immune cells surrounding the airway and the occlusion of vessels with immune cells. Ten medium to large vessels were scored per sample and averaged to generate a score per animal. 

### 4.10. Immunofluorescence and Microscopy

FFPE slides were deparaffinized by xylene, rehydrated in decreasing percent ethanol solutions, followed by heat-induced epitope retrieval with a Tris-EDTA solution for 20 min (10 mM Tris, 1 mM EDTA, 0.5% Tween-20, pH 9.0, heated at 100 °C for 20 min). Slides were then treated with blocking buffer for 1 h and stained with anti-YAP (1:500, rabbit polyclonal, 13584-1-AP, Proteintech Group, Rosemont, IL, USA) or anti-TAZ (1:500, rabbit polyclonal, 23306-1-AP, Proteintech Group) antibodies diluted in blocking buffer overnight at 4 °C. Blocking buffer consisted of 1% bovine serum albumin (ThermoFisher Scientific, Waltham, MA, USA), 0.2% cold-water fish gelatin (Sigma-Aldrich, St. Louis, MO, USA), 0.1% Tween-20 (ThermoFisher Scientific), and 0.1% sodium azide (Sigma-Aldrich) in PBS. Slides were washed three times in PBS and stained with Cy5-conjugated goat anti-rabbit (1:500, 072021506, KPL) diluted in blocking buffer for 2 h and counterstained with DAPI in mounting media (Vector Laboratories VECTASHIELD Antifade Mounting Medium with DAPI, UX-93952-24). Lung sections were imaged on a Leica DMI8 automated microscope using a DFC9000 sCMOS camera and a 20× Plan Apochromat objective (Leica, Buffalo Grove, IL, USA). For each tissue, a total area of approximately 5.4 mm^2^ was imaged as a tilescan. Each tilescan was stitched into a single image using a custom MATLAB (MATLAB 2021a, Mathworks, Natick, MA, USA) implementation of the Phase Correlation Method previously described [102]. Images were analyzed for YAP and TAZ expression using a semi-automated segmentation. Briefly, a blinded observer manually annotated images to remove large artifacts or non-lung tissue. Following this, the DAPI channel was filtered using a 3 × 3 median filter (1.95 μm × 1.95 µm) and a 9.75 μm radius rolling ball filter. A nuclear mask was then segmented from the background using global binary thresholding. Each image and segmentation were manually checked by a blinded observer. The fluorescent staining intensity of YAP and TAZ was averaged over the nuclear mask. To remove variable background from the imaging, background subtraction was performed with a 650 μm radius rolling ball filter.

### 4.11. Statistical Analyses

Statistical analyses were performed using Graphpad Prism Version 9.4.1. For bar graphs with error bars, the standard error of the mean (SEM) was used. Ordinary two- and three-way ANOVA analyses were fit to a full model, followed by Tukey’s post hoc test, and reported with multiplicity-adjusted *p*-values. Mann–Whitney analyses were used for intestinal worm burden data. The α-cutoff was set at 0.05 for all analyses.

## 5. Conclusions

Using wild-type and *Ffar4*^−/−^ mice in pulmonary exposures, we identified a role for FFAR4-signaling, independent of exogenous DHA ligation, in eliciting an immune response and epithelial homeostasis, possibly through cytoprotection and innate/adaptive immunity liaising. 

## Figures and Tables

**Figure 1 ijms-24-07072-f001:**
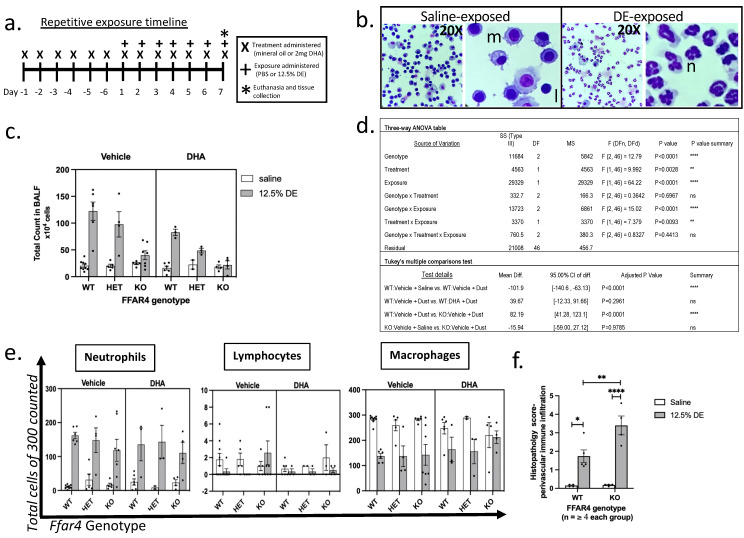
Significantly dampened immune recruitment in *Ffar4*−/− following repetitive pulmonary dust exposure with no changes to cell populations. (**a**) A schematic of the experimental timeline of repetitive exposure with treatment. Mice were given a 6-day pre-treatment of mineral oil or DHA by oral gavage, followed by a 7-day concurrent treatment with exposure to saline or DE. Mice were euthanized 5 h after the final doses on day 7 of the exposure period. (**b**) Representative DiffQuick-stained cytospins of BAL cells for saline- and DE-exposed mice show macrophages (m) and neutrophils (n) in the airways (20× and 40×). Repetitive DE exposure over 7 days significantly (three-way ANOVA followed by Tukey’s post hoc test) induces BALF cell influx in an exposure- and genotype-dependent manner (**c**,**d**). This influx is marked by neutrophilia in DiffQuick-stained-cytospins determined by cell differential analysis from masked samples (**e**). There were no significant changes in cell differentials between genotypes, though exposure induced neutrophilia (main effect of exposure: *p* < 0.0001). (**f**) H & E stained FFPE lung sections were masked and scored for vascular inflammation. Inflammation was induced by DE exposure in both WT and KO compared with their genotype-matched controls (Adj. *p* = 0.0127 and <0.0001 respectively). *Ffar4* KO mice had significantly higher inflammatory scores than WT (Adj. *p* = 0.0095). Experiments were performed in triplicate (at minimum) with all groups represented in each separate experiment. * *p* < 0.05; ** *p* < 0.01; **** *p* < 0.0001.

**Figure 2 ijms-24-07072-f002:**
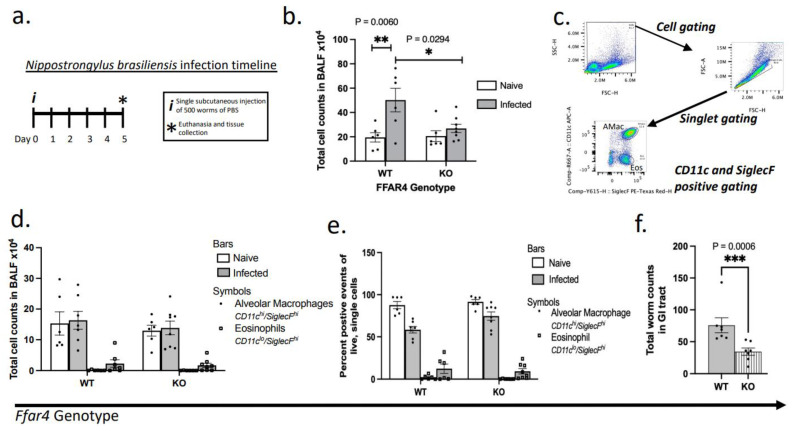
The *Ffar4*−/− phenotype of dampened immune influx without changes to cell populations is confirmed in another model: Nippostrongylus brasiliensis (Nb) infection. (**a**) A schematic of the helminth infection timeline. Mice were subcutaneously injected with 500 Nippostrongylus brasiliensis L3 larvae in 200 μL PBS or PBS only on day 0. Mice were euthanized on day 5 post-infection. (**b**) Helminth-infected WT mice had significantly elevated cell counts in the airway compared with their naïve controls, as measured by total cell counts in the BALF; this effect was significantly dampened by *Ffar4*-deficiency, being comparable to naïve controls. (**c**–**e**) BAL cell differentials, determined by flow cytometry, were not significantly altered by *Ffar4* presence, though the infection significantly impacted cell differentials by promoting eosinophilia in the airways (two-way ANOVA, main effect *p* < 0.0001). Expressing the data as totals or percentages did not alter this outcome. These data were generated from two separate experiments with representative groups included in each experiment. (**f**) *Ffar4*-deficiency dampened the presence of worms in the GI tract, with significantly fewer worms counted from the jejunum of *Ffar4*-null mice. * *p* < 0.05; ** *p* < 0.01; *** *p* < 0.001.

**Figure 3 ijms-24-07072-f003:**
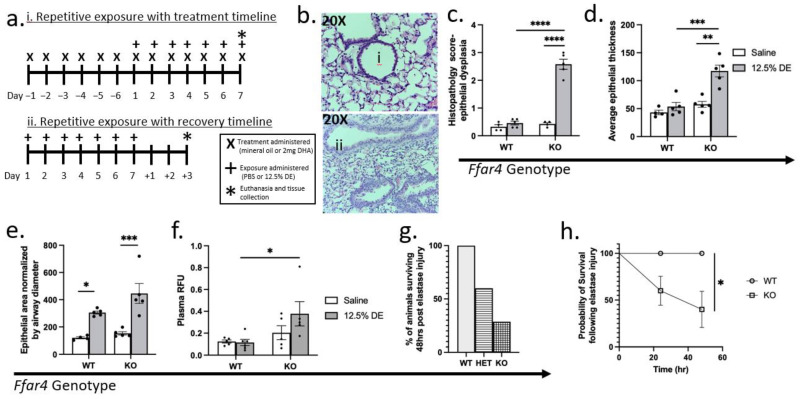
Lack of *Ffar4* impairs airway epithelial barrier homeostasis. (**a**) Schematic of repetitive exposure timeline without (**i**) and with (**ii**) the recovery period. The recovery studies did not include treatments, and mice were left to recover without exposure for three days. On the third day of recovery, 1 h before euthanizing, mice were given intranasal RhoB-dextran in PBS. X’s represent oral gavage treatments with mineral oil (vehicle control) or 2 mg DHA, plus signs (+) represent the administration of intranasal exposure (PBS or 12.5% DE), and the asterisk (*) signifies euthanasia. If euthanasia fell on the day of an exposure, animals were euthanized 5 h after administration. (**b**) H & E-stained FFPE lung sections reveal a dysplastic epithelial appearance (representative images of (**i**) normal and (**ii**) dysplastic epithelium at 20×). *Ffar4*-deficient mice exposed to DE had significantly higher airway epithelial dysplasia scores compared with DE-exposed WT and naïve KO mice ((**c**), Adj. *p* < 0.0001 for each), and similar findings were observed in quantified epithelial thickness ((**d**), Adj. *p* = 0.0001 and 0.0002, respectively). Airway area normalized to diameter was only significant in both dust-exposed WT and KO mice compared with their respective genotype-matched saline controls ((**e**), Adj. *p* = 0.0285 and 0.004 respectively). (**f**) Using intranasal Rhodamine B dextran delivery, *Ffar4*-deficient mice had significantly greater plasma fluorescence following repetitive instillations of DE plus three days for repair (Adj. *p* = 0.0081). (**g**,**h**) *Ffar4*-deficient mice had significantly reduced survival in an acute epithelial injury model of a single dose of intranasal porcine pancreatic elastase ((**h**), Log-rank test, *p* = 0.0224). ** *p* < 0.01; **** *p* < 0.001; **** *p* < 0.0001.

**Figure 4 ijms-24-07072-f004:**
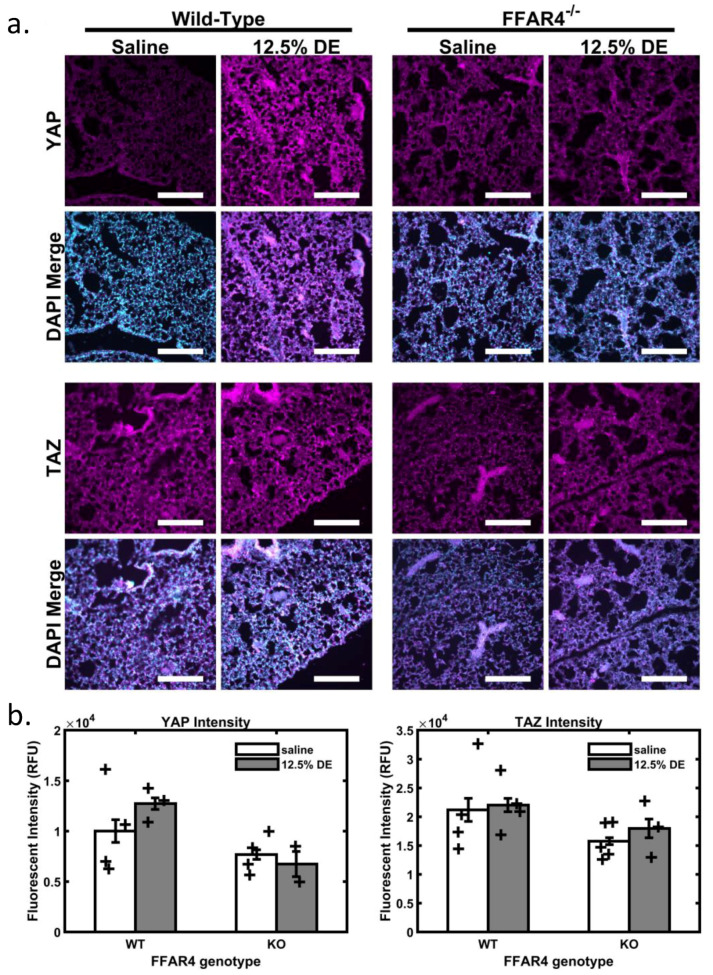
Lack of *Ffar4* and reduced YAP signal in lung immunofluorescence. Immunofluorescence was performed on FFPE lung sections from mice repetitively exposed to 12.5% DE or saline (5 μm sections imaged at 20×; scale bars are 200 μm). (**a**) Anti-murine YAP or TAZ antibodies were used, followed by secondary antibody incubation, then counter-stained with DAPI. (**b**) YAP intensity showed a significant main-effect of genotype (2-way ANOVA, *p* = 0.0424), while TAZ was not significant in these comparisons.

**Figure 5 ijms-24-07072-f005:**
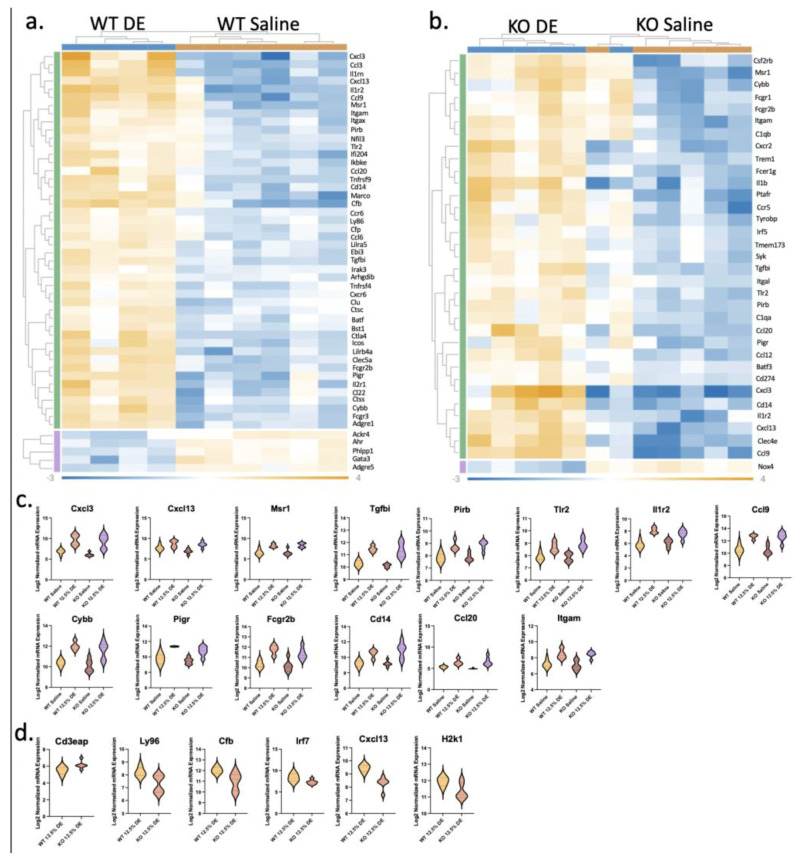
Dust-induced differentially expressed genes in total lung RNA from repetitively exposed mice. Total lung bulk RNA from mice repetitively exposed to 12.5% DE or saline was analyzed by the mouse immunology NanoString nCounter panel. Significant genes (Adj. *p* ≤ 0.05) with a fold-change of +/−1.5× are reported as heat maps of differentially expressed genes (DEG) in DE-exposed WT mice compared with WT saline controls ((**a**), 51 genes) or DE-exposed KO mice compared with KO saline controls ((**b**), 34 genes). One of the KO DE samples clusters with the saline group, which could be an artifact of poor intranasal delivery. Violin plots of the (**c**) 14 DEGs overlapping in both DE-exposed WT and KO samples compared with their genotype-matched saline controls (Adj. *p* < 0.05, FC: +/−1.5×) and (**d**) 5 DEGs between DE-exposed KO compared with WT (*p* < 0.05, FC: +/−1.5×).

**Table 1 ijms-24-07072-t001:** Summary of pathway analysis findings using significant DEGs. Differentially expressed genes induced by DE exposure which were unique in WT (upper section) or KO (middle section) compared with their genotype controls were input into REACTOME, an open-source pathway database. Only genes that were unique to WT or KO with an Adj. *p* < 0.05 and fold-change of +/−1.5× were used. (lower section) DEGs in DE-exposed KO compared with WT samples with *p* < 0.05 and a fold-change of +/−1.5× were input into REACTOME. Only significant pathways with a *p*-value < 0.05 and FDR < 0.1 are reported in this table.

Pathway Analysis of Unique DEGs from WT DE-Exposed vs. WT Saline Comparison	
Reactome Pathway Name	*p*-Value
RUNX1 and FOXP3 control the development of Tregs	5.13 × 10^−5^
Chemokine receptors bind chemokines	6.03 × 10^−5^
Interleukin-10 signaling	5.40 × 10^−4^
Alternative complement activation	2.24 × 10^−3^
Activation of C3 and C5	2.24 × 10^−3^
Interleukin-4 and Interleukin-13 signaling	3.38 × 10^−3^
**Pathway Analysis of Unique DEGs from KO DE-Exposed vs. KO Saline Comparison**	
**Reactome Pathway Name**	***p*-Value**
CLEC7A/inflammasome pathways	2.55 × 10^−6^
Interleukin-10 signaling	2.74 × 10^−5^
Interleukin-1 processing	3.06 × 10^−5^
Pyroptosis	4.78 × 10^−4^
C-type lectin receptors	1.04 × 10^−3^
RUNX3 regulates immune response and cell migration	2.12 × 10^−3^
Regulated necrosis	4.76 × 10^−3^
Dectin-2 family	7.14 × 10^−3^
**Pathway Analysis of DEGs in DE-Exposed KO vs. WT Mice**	
**Reactome Pathway Name**	***p*-Value**
Activation of IRF3/IRF7-mediated TBK1/IKK epsilon	4.66 × 10^−4^
TRAF6-mediated IRF7 activation of TLR7/8 or 9 signaling	2.21 × 10^−3^
TICAM1-dependent activation of IRF3/IRF7	4.42 × 10^−3^
TRAF6-mediated IRF7 activation	8.81 × 10^−3^
TRAF3-dependent IRF activation pathway	8.81 × 10^−3^
Alternative complement activation	1.32 × 10^−2^
Activation of C3 and C5	1.32 × 10^−2^
DEx/H-box helicases activate type I IFN and inflammatory cytokine production	1.53 × 10^−2^
MyD88-dependent cascade initiated on endosome	1.94 × 10^−2^
Toll Like receptor 7/8 (TLR7/8) cascade	1.98 × 10^−2^
TRIF-mediated programmed cell death	2.19 × 10^−2^
Toll-like receptor 9 cascade	2.25 × 10^−2^
MyD88-independent TLR4 cascade	2.34 × 10^−2^
TRIF(TICAM1)-mediated TLR4 signaling	2.34 × 10^−2^
TRIF-mediated programmed cell death	2.41 × 10^−2^
Toll-like receptor 4 cascade	3.24 × 10^−2^
Activation of IRF3/IRF7 mediated by TBK1/IKK epsilon	3.35 × 10^−2^
Caspase activation via Death Receptors in the presence of ligand	3.48 × 10^−2^
Activation of TAK1 complex upon TLR7/8 or 9 stimulation	3.82 × 10^−2^
TRAF6-mediated induction of TAK1 complex within TLR4 complex	4.00 × 10^−2^
IRAK deficiency (TLR2/4)	4.17 × 10^−2^
IKK complex recruitment mediated by RIP1	4.47 × 10^−2^
Caspase activation via extrinsic apoptotic signaling pathway	4.69 × 10^−2^
Heme signaling	4.91 × 10^−2^

## Data Availability

NanoString data files will be provided to any interested individuals; please reach out to the authors for these files.

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
