# Peer review of "Lipid-Sensing Receptor FFAR4 Modulates Pulmonary Epithelial Homeostasis following Immunogenic Exposures Independently of the FFAR4 Ligand Docosahexaenoic Acid (DHA)"

_ijms, 2023, doi:10.3390/ijms24087072_

Round 1

Reviewer 1 Report

In the manuscript “Lipid-sensing receptor FFAR4 modulates pulmonary epithelial 2 homeostasis following immunogenic exposures independently 3 of FFAR4 ligand docosahexaenoic acid (DHA)” Sveiven and collaborations researched the role of FFAR4 in the pulmonary immune response in a in vivo model using DE as immunogenic stimulus. Also, they postulate the Hippo pathway as a potential mechanism involved. The experimental design is well conducted, and conclusions are supported by results, however some points should be addressed by authors to improve the quality of manuscript.

1.       The introduction is long, several topics are described and look disconnected. Please, revise this section and include the necessary background to get the objective of this work.

2.       The quality of figures should be improved. For example, the figure 1 b and the table 1d doesn’t have a good resolution.

3.       The microscopy images in figures 1, 3 and 4 should include the magnifications.

4.       The discussion section is too long. This information should be focus on discussing the experimental findings including the necessary literature.

5.       References included in the manuscript are excessive. A lot of information can be obvious for the authors but not for a general reader that want to have support from the value of the knowledge claimed in this work.

6.       In supplementary material, table S1 and video S1 are missing.

Author Response

In the manuscript “Lipid-sensing receptor FFAR4 modulates pulmonary epithelial 2 homeostasis following immunogenic exposures independently 3 of FFAR4 ligand docosahexaenoic acid (DHA)” Sveiven and collaborations researched the role of FFAR4 in the pulmonary immune response in a in vivo model using DE as immunogenic stimulus. Also, they postulate the Hippo pathway as a potential mechanism involved. The experimental design is well conducted, and conclusions are supported by results, however some points should be addressed by authors to improve the quality of manuscript.

  1. The introduction is long, several topics are described and look disconnected. Please, revise this section and include the necessary background to get the objective of this work. Response: We have taken this opportunity to make the introduction more cohesive to address this comment.
  2. The quality of figures should be improved. For example, the figure 1 b and the table 1d doesn’t have a good resolution. Response: Thank you for identifying these poor quality images and we have provided better images as suggested.
  3. The microscopy images in figures 1, 3 and 4 should include the magnifications. Response: For clarity we have added the magnifications for these images, thank you.
  4. The discussion section is too long. This information should be focus on discussing the experimental findings including the necessary literature. Response: The manuscript has been updated to consolidate the discussion to only that which pertains to the literature and the findings of this work.
  5. References included in the manuscript are excessive. A lot of information can be obvious for the authors but not for a general reader that want to have support from the value of the knowledge claimed in this work. Response: We have reviewed and revised the manuscript as appropriate to increase clarity and to ensure appropriate discussion and use of citations/references throughout.
  6. In supplementary material, table S1 and video S1 are missing. Response: Thank you for pointing out this error, we have updated the revised manuscript to include only the supplementary files we are including.

Reviewer 2 Report

The main findings of this study are that DHA mediates anti-inflammatory effects independent of and that dust from swine confinement facilities (DE)-exposed mice lacking FFAR4 had reduced immune cells in the airways, epithelial dysplasia, and impaired pulmonary barrier integrity.

The paper has been written in a very clear style and the content appears to be coherent.

  1. Why do the authors focus on anti-inflammatory effects of DHA and not e.g. EPA? EPA and DHA do have quite distinct effects e.g. in cardiovascular outcome trials.
  2. Much is already known on the phenotype of Ffar4-null mice. Are these results fundamentally new?
  3. Ffar4-null mice and wild-type mice were generated by crossing heterozygous mice and littermates were compared in these experiments. What is the genetic background?

Author Response

The main findings of this study are that DHA mediates anti-inflammatory effects independent of and that dust from swine confinement facilities (DE)-exposed mice lacking FFAR4 had reduced immune cells in the airways, epithelial dysplasia, and impaired pulmonary barrier integrity.

The paper has been written in a very clear style and the content appears to be coherent.

  1. Why do the authors focus on anti-inflammatory effects of DHA and not e.g. EPA? EPA and DHA do have quite distinct effects e.g. in cardiovascular outcome trials. Response: We added a statement highlighting that many studies suggest DHA in particular mediates a broader range of anti-inflammatory effects (10.1016/j.atherosclerosis.2020.11.018). We feel providing this clarity was certainly warranted, and are thankful this comment was made.
  2. Much is already known on the phenotype of Ffar4-null mice. Are these results fundamentally new? Response: We have added a statement to the introduction to clarify the gap that is addressed by this work particularly as it pertains to the lung. Thank you for shedding light on this gap.
  3. Ffar4-null mice and wild-type mice were generated by crossing heterozygous mice and littermates were compared in these experiments. What is the genetic background? Response: We added a statement to clarify this as we agree with the reviewer on it’s importance to state clearly in the Ffar4 knockout mouse models section of the methods.